# c-Abl Regulates the Pathological Deposition of TDP-43 via Tyrosine 43 Phosphorylation

**DOI:** 10.3390/cells11243972

**Published:** 2022-12-08

**Authors:** Saebom Lee, Hye Guk Ryu, Sin Ho Kweon, Hyerynn Kim, Hyeonwoo Park, Kyung-Ha Lee, Sang-Min Jang, Chan Hyun Na, Sangjune Kim, Han Seok Ko

**Affiliations:** 1Neuroregeneration and Stem Cell Programs, Institute for Cell Engineering, Johns Hopkins University School of Medicine, Baltimore, MD 21205, USA; 2Department of Neurology, Johns Hopkins University School of Medicine, Baltimore, MD 21205, USA; 3Department of Biological Sciences, Korea Advanced Institute of Science and Technology, Daejeon 34141, Republic of Korea; 4Department of Biological Sciences and Biotechnology, Chungbuk National University, Cheongju 28644, Republic of Korea; 5Department of Cosmetic Science and Technology, Daegu Haany University, Gyeongsan 38610, Republic of Korea; 6Department of Molecular Biology, Pusan National University, Busan 46241, Republic of Korea; 7Department of Biochemistry, Chungbuk National University, Cheongju 28644, Republic of Korea

**Keywords:** c-Abl, TDP-43, phosphorylation, mislocalization, amyotrophic lateral sclerosis

## Abstract

Non-receptor tyrosine kinase, c-Abl plays a role in the pathogenesis of several neurodegenerative disorders such as Alzheimer’s disease and Parkinson’s disease. Here, we found that TDP-43, which was one of the main proteins comprising pathological deposits in amyotrophic lateral sclerosis (ALS), is a novel substrate for c-Abl. The phosphorylation of tyrosine 43 of TDP-43 by c-Abl led to increased TDP-43 levels in the cytoplasm and increased the formation of G3BP1-positive stress granules in SH-SY5Y cells. The kinase-dead mutant of c-Abl had no effect on the cytoplasmic localization of TDP-43. The expression of phosphor-mimetic mutant Y43E of TDP-43 in primary cortical neurons accumulated the neurite granule. Furthermore, the phosphorylation of TDP-43 at tyrosine 43 by c-Abl promoted the aggregation of TDP-43 and increased neuronal cell death in primary cortical neurons, but not in c-Abl–deficient primary cortical neurons. Identification of c-Abl as the kinase of TDP43 provides new insight into the pathogenesis of ALS.

## 1. Introduction

Amyotrophic lateral Sclerosis (ALS) has characterized a progressive neurodegenerative disease that affects motor neurons in the brain and spinal cord [1]. ALS-affected neurons and glia showed an abnormal aggregation of TAR DNA-binding protein 43 (TDP-43), which is one of pathological hallmark of ALS and multiple forms of frontotemporal lobar degeneration (FTLD) [2]. Accumulating evidence has reported that rare mutations in *TARDBP*, the gene encoding TDP-43, are dominantly inherited in familiar ALS patients, and are found in individuals with sporadic ALS [3,4], which support that TDP-43 dysfunction leads to neurodegeneration.

RNA binding protein TDP-43 is mainly located in the nucleus due to its nuclear localization signal (NLS) [5], and plays an important role in RNA processing through interacting with ribonucleoprotein complexes to control pre-RNA maturation [6]. One of features in the pathogenesis of ALS is the mislocalization of TDP-43, leading to loss of nuclear TDP-43 and gain additional roles due to the subcellular localization of TDP-43 [7]. Loss of functions of nuclear TDP-43 have deleterious effects on the splicing of target RNAs [8], but also gain of functions of cytoplasmic TDP-43 regulate the formation of stress granule and the sequestration of essential mRNA transcripts [9,10]. Moreover, the cytoplasmic accumulation of TDP-43 may be prone to expose to protease, resulting in the carboxy-terminal fragments (CTFs) of TDP-43, which are enriched in the cytoplasmic inclusion within ALS-affected neurons [9]. Together with proteolytic cleavage, aberrant post-translational modification of TDP-43 was frequently observed in ALS-affected neurons, suggesting this modification could affect the cellular location and functions [11]. Specifically, TDP-43 pathology appears to be tightly linked with hyperphosphorylation of TDP-43, making it into cytoplasmic inclusion and seems to be toxic to neurons [12,13,14].

Since the aberrant regulation of kinases has been proposed as one of the pathological drivers in selective motor neuron degeneration, many studies have reported that several kinase inhibitors have beneficial effects in multiple models of ALS [15]. The non-receptor tyrosine kinase, c-Abl, was increased in post-mortem spinal cord tissue from sporadic ALS patients [16]. Moreover, compounds related c-Abl inhibition have been linked to motor neuron survival assessed by a compound screening of induced pluripotent stem cell (iPSC) derived motor neurons from ALS patients [17], suggesting c-Abl functions in the pathogenesis of ALS. Consistent with this notion, c-Abl inhibition has shown to alleviate the motor neuron degeneration in the iPSC-derived motor neurons from familial and sporadic cases, and in mutant SOD1 and TDP-43 transgenic mice [17,18,19]. However, little is known about the role of c-Abl activation in the pathogenesis of ALS. Here, we show that the c-Abl kinase directly phosphorylates TDP-43, promoting its cytoplasmic location and accumulation. In addition, we find that c-Abl–mediated TDP-43 phosphorylation increases the formation of cytoplasmic stress granules and insoluble TDP-43 levels in both full-length and CTFs, leading to neurodegeneration.

## 2. Materials and Methods

### 2.1. In Vitro Kinase Assay

Kinase assays utilizing 1 μg recombinant c-Abl (active, Sigma-Aldrich, St. Louis, MO, USA) and 1 μg recombinant proteins (Tau or TDP-43 purchased from R&D systems, Minneapolis, MN, USA, or α-synuclein purified in lab) as substrates were performed at 30 °C in 20 μL reaction mixture containing kinase buffer (10 mM Tris-HCl [pH 7.5], 10 mM MgCl_2_, 1 mM dithiothreitol, and 32P-γ-ATP). After 30 min, reactions were resolved by sodium dodecyl sulfate (SDS) polyacrylamide gel electrophoresis (PAGE) and visualized with autoradiography.

### 2.2. Cell Culture and Transfection

SH-SY5Y cells were grown in Dulbecco’s Modified Eagle Medium (DMEM) supplemented with 10% fetal bovine serum (FBS) and 100 U/mL each of penicillin and streptomycin (Gibco, Grand Island, NY, USA). Transient transfection was carried out with Lipofectamine LTX plus reagents (Invitrogen, Carlsbad, CA, USA) according to manufacturer’s protocols.

### 2.3. Primary Cortical Neuron Culture and Lenti-Viral Transduction

To obtain c-Abl deficient neurons, the floxed c-Abl mice were mated with nestin-Cre expressing mice as described previously [20]. Primary cortical neurons were prepared from E14.5 pups and cultured in Neurobasal media supplemented with B-27, 0.5 mM L-glutamine, 100 U/mL each of penicillin and streptomycin (Gibco) on 24-well tissue culture plates coated with poly-D-lysine (Sigma-Aldrich). For expression of TDP-43 mutants in the primary cortical neurons, the lenti-viruses were prepared from HEK293FT transfected with cFUGW-TDP-43 WT or mutants together viral packaging plasmids purchased from Invitrogen. Infectious lenti-viruses were harvested after 48 h post-transfection. The supernatant was collected, filtered, and concentrated by ultracentrifugation at 25,000× *g* for 3 h. The pellet was resuspended in 1% BSA in PBS, and then stored at −80 °C until use. Days in vitro (DIV) 7 neurons were transduced with control or LV-TDP-43 WT or LV-TDP-43 mutants with a multiplicity of infection (MOI) of 5.

### 2.4. Immunoprecipitation

EGFP-tagged c-Abl expressing vector was co-transfected with Flag-tagged TDP-43 WT or mutants into SH-SY5Y cells. After 24 h post-transfection, cell extracts were incubated with rotation with the anti-Flag (M2, Sigma) antibody at 4 °C for 1 h and then further incubation with protein G-agarose beads (Roche, Mannheim, Germany) for overnight. Immunoprecipitates were washed five times with lysis buffer and subjected to immunoblotting.

### 2.5. Molecular Docking Study

PatchDock, which is a freely available web server, was used to study the structural relationship between TDP-43 and c-Abl. According to the crystal structure of TDP-43 (PDB entry: 5MDI) [21] and c-Abl (PDB entry: 2E2B) [22], the best structure for docking was selected and further analyzed using PyMOL.

### 2.6. Subcellular Fractionation

Cytoplasmic and nuclear lysates were prepared from SH-SY5Y cells by lysis in a buffer containing 10 mM HEPES (pH 7.6), 3 mM MgCl_2_, 40 mM KCl, 2 mM DTT, 5% glycerol, 0.5% NP-40, 0.5 mM PMSF and protease/phosphatase inhibitor cocktail (Roche) at 4 °C. Nuclei were removed by centrifuge at 1200× *g* for 5 min. Nuclear lysates were then prepared from the pelleted nuclei in a buffer containing 10 mM HEPES (pH 7.9), 0.1 mM EGTA, 1.5 mM MgCl_2_, 420 mM NaCl, 0.5 mM DTT, 0.5 mM PMSF, 25% glycerol and protease/phosphatase inhibitor cocktail (Roche). After 30 min incubation in ice, samples were subjected to sonication for 15 s at 10% amplitude. Nuclear lysates were obtained in the supernatant by centrifuge at 22,000× *g* at 4 °C for 30 min. Soluble and insoluble fraction were prepared from TDP-43 WT or mutant expressing primary cortical neurons by lysis in serial buffers. For soluble fraction, cells were incubated in a buffer containing 10 mM Tris (pH 7.5), 150 mM NaCl, 1% Triton X-100 and protease/phosphatase inhibitors for 30 min in ice. After sonication, the cell lysates were centrifuged at 22,000× *g* at 4 °C for 30 min. Pellets were solubilized in 8 M urea buffer containing protease/phosphatase inhibitors.

### 2.7. Western Blotting

Cell lysates were determined the protein concentration using the BCA assay (Pierce, Rockford, IL, USA) and 10 μg protein samples were separated on 8–16% gradient SDS-PAGE gels (Life technologies, Carlsbad, CA, USA) and transferred to nitrocellulose membrane (0.45 μm pore size, Bio-Rad). After 30 min incubation with 5% skim milk in TTBS buffer (10 mM Tris (pH7.5), 150 mM NaCl, 0.05% Tween-20) for blocking, membrane was incubated at 4 °C overnight with primary antibodies; anti-Flag (1:1000, Sigma, Cat#F1804), anti-GFP (1:2000, Abcam, Cat#ab290), anti-TDP-43 (1:1000, Proteintech, Cat#12892-1-AP), anti-Lamin B (1:1000, Abcam, Cat#ab16048), anti-GAPDH (1:1000, Santa-Cruz, Cat#sc-32233), and anti-c-Abl (1:1000, CST, Cat#2862) antibodies at 4 °C overnight followed by incubation with HRP-conjugated rabbit of mouse secondary antibodies (1:50,000; GE Healthcare) and HRP-conjugated mouse of donkey secondary antibodies (1:10,000; GE Healthcare) for 1 h at room temperature (RT). Immunoblot signals were visualized by enhanced chemiluminescence (Thermo Scientific, Rockford, IL, USA). The membranes were reprobed with HRP-conjugated anti-β-actin antibody (1:40,000, Sigma, Cat#A3854).

### 2.8. Cell Death Assay

For detection and quantification of apoptosis, TDP-43 WT and phospho-mimetic mutants expressing primary cortical neurons were assessed by In Situ Cell Death detection kit (Roche) according to manufacturer’s protocol. Briefly, neurons were subjected to lenti-viral transduction at DIV 5. After 14 days transduction, neurons were fixed in 4% PFA for 15 min, followed by 3 times washing. Neurons were then permeabilized in a buffer containing 0.1% Triton X-100 for 20 min. After washing, TUNEL reaction mixture added to neurons and incubated for 1 h at 37 °C in the dark. Following 3 times washing, the samples were analyzed under a fluorescence microscope.

### 2.9. Fluorescence Microscopy

SH-SY5Y cells were grown onto coverslips until they reached 50–60% confluence. Transfected cells were maintained for 24 h, fixed with 4% paraformaldehyde for 15 min. For the formation of stress granule, SH-SY5Y cells were treated with 0.5 mM Sodium arsenite for 30 min prior to fixation. After permeabilization with 0.2% Triton X-100 for 20 min, and if necessary, immunostained with primary antibody against G3BP1 (1:500, Proteintech, Cat#13057-2-AP) and fluorescent-conjugated secondary antibodies. For primary cortical neurons, we utilized cover slips with poly-D-lysine coating to plate the cells. A total of 4% paraformaldehyde was used to fix the cells, followed by blockage in a solution with 5% normal donkey serum (Jackson Immunoresearch, West Grove, PA, USA), and 0.1% Triton X-100 for 1 h at room temperature. After a series of incubations with primary antibodies at 4 °C overnight, the samples were washed with 0.1% Triton X-100 in PBS, followed by 1 h of incubation of the coverslips with a mixture of FITC-conjugated (Jackson Immunoreserach) and CY3-conjugated (Jackson Immunoreserach) secondary antibodies at room temperature. The fluorescent imagines were acquired via Zeiss confocal microscope (Zeiss Confocal LSM 710) after the coverslips were mounted. All images were processed by the Zeiss Zen software. The selected area in the signal intensity range of the threshold was measured using ImageJ analysis.

### 2.10. Filter Trap Assay

Insoluble fraction of cell lysates was diluted in 1% SDS-PBS and boiled for 5 min. Immediately after cooling, the serial dilutions of proteins were loaded onto nitrocellulose membranes (0.2 μm pore size) settled on a dot blotter (Bio-Rad, Hercules, CA, USA). After blocking with 5% skim milk in TBST for 1 h, the membranes were incubated with anti-TDP-43 and anti-β-actin antibodies for overnight. HRP activity was detected by developing in SuperSignal™ West Pico PLUS Chemiluminescent Substrate (Thermo Fisher Scientific, Waltham, MA, USA).

### 2.11. Mass Spectrometry Analysis

The excised SDS-PAGE bands were diced into ~1 mm-sized cubes with a clean scalpel, followed by washing with 40% acetonitrile (ACN) and 50 mM triethylammonium bicarbonate (TEAB) until all the gel pieces became clear. The proteins in the gel pieces were reduced and alkylated by incubating them in 10 mM dithiothreitol at 60 °C for 1 h and subsequently in 30 mM iodoacetamide at room temperature for 15 min. After washing the gel pieces with water for 10 min, they were dehydrated in ACN and vacuum-dried. The proteins in the gel pieces were then digested by rehydrating them with 10 ng/µL of trypsin (sequencing grade modified trypsin; Promega, Fitchburg, WI, USA) in 20 mM TEAB and incubating them at 37 °C overnight. The peptides trapped inside the gel pieces were extracted with 50% ACN/0.1% formic acid (FA) thrice and ACN once. The peptides were desalted with C18 Stage-Tips (3M EmporeTM; 3M, St. Paul, MN, USA). The eluted peptide solution was vacuum-dried and then reconstituted in 0.1% FA. For mass spectrometry analysis, the peptides were trapped onto an Acclaim™ PepMap™ 100 LC C18 NanoViper trap column (100 μm × 2 cm, packed with 5-μm C18 particles, Thermo Scientific) at a flow rate of 10 μL/min and resolved on an EASY-Spray™ analytical column (75 μm × 50 cm, packed with 2-μm C18 particles, Thermo Scientific) at a flow rate of 0.25 μL/min using an EASY-nLC 1200 nanoflow liquid chromatography system (Thermo Scientific) that was coupled with an Orbitrap Fusion Lumos Tribrid Mass Spectrometer. The peptide separation was conducted by increasing the gradient of solvent B (0.1% FA in 95% ACN) from 5% to 12% for 2 min, from 12% to 25% for 30 min, from 25% to 35% for 15 min, and from 35% to 95% for 2 min. An EASY-Spray ion source was operated at 2.3 kV. The data acquisition for the peptides injected into the mass spectrometer was conducted in data-dependent acquisition (DDA) mode. The MS1 scan range was set to *m*/*z* 350 to 1550 with 3-s per cycle of the “top speed” setting. The mass resolutions for MS1 and MS2 were 120,000 and 30,000 at an *m*/*z* of 200, respectively. Maximum ion injection times for MS1 and MS2 were 50 and 100 milliseconds, respectively. Automatic gain controls for MS1 and MS2 were 1 and 0.05 million ions, respectively. Higher-energy collisional dissociation value was set to 32%. The precursor isolation window was set to *m*/*z* 1.6 with an *m*/*z* 0.3 of offset. Dynamic exclusion was set to 30 s with 10 ppm of the mass window. Singly charged ions were rejected. Internal calibration was conducted using the lock mass option (*m*/*z* 445.1200025) from ambient air [23,24,25]. The identification of peptides was conducted using MaxQuant (version 1.5.3.8) software [26]. The tandem mass spectrometry data were then searched using Andromeda algorithms against a human TARDBP protein sequence (NP_031401.1 from the RefSeq database) with common contaminant proteins (245 entries) embedded in MaxQuant software. The search parameters used were as follows: (a) trypsin as a proteolytic enzyme with up to 2 missed cleavages; (b) first search peptide tolerance of 20 ppm; (c) main search peptide tolerance of 4.5 ppm; (d) fragment mass error tolerance of 20 ppm; and (e) carbamidomethylation of cysteine (+57.02146 Da) as fixed modifications; (f) oxidation (+15.99492 Da) of methionine, acetyl on protein N-terminal (+42.01057 Da) and phosphorylation (+79.96633 Da) on serine, threonine, and tyrosine as variable modifications. The minimum peptide length was set to 6 amino acids and the minimum number of peptides per protein was set to 1. Peptides and proteins were filtered at a 1% false-discovery rate (FDR) [23,24,25].

### 2.12. Statistical Analysis

All data were analyzed using GraphPad software (version 6) and were presented as mean ± standard error of the mean (SEM) with at least three independent experiments. One-way ANOVA with Tukey’s post hoc test was used for immunostaining quantification.

## 3. Results

### 3.1. c-Abl Phosphorylates TDP-43

Since c-Abl inhibition in multiple models of ALS shows therapeutic effects [17,18,19], we tried to test whether c-Abl could post-translational regulate TDP-43, which is one of aggregation-prone proteins in ALS. In order to confirm that c-Abl phosphorylates TDP-43 directly, we performed in vitro kinase assays using purified recombinant TDP-43 proteins. We used alpha-synuclein and Tau proteins as positive controls, which were reported as substrates of c-Abl [27,28,29]. We found the phosphorylated TDP-43 depends on active c-Abl protein (Figure 1A). To determine the c-Abl–specific phosphorylation sites of TDP-43, we resolved TDP-43 by SDS-PAGE after the incubation with active c-Abl and cropped the shifted TDP-43 band (Figure 1B). MALDI-TOF analyses revealed that c-Abl kinase phosphorylated TDP-43 at tyrosine 43 residue (Y43) (Figure 1C). To further confirm that c-Abl phosphorylates TDP-43 at Y43, we constructed a Y43F phosphor-deficient mutant of TDP-43 as well as three different Y73F, Y155F, and Y374F phosphor-deficient mutants of TDP-43 as negative controls. Co-transfection and immunoprecipitation experiments in SH-SY5Y cells revealed that the phosphorylated levels of TDP-43 Y73F, Y155F, and Y374F phosphor-deficient mutants by c-Abl were similar to the phosphorylated levels of TDP-43 WT, but the phosphorylated levels of TDP-43 Y43F by c-Abl was significantly reduced (Figure 1D). The results indicate that c-Abl specifically phosphorylates the TDP-43 at Y43 residue. 

### 3.2. c-Abl Interacts with TDP-43

In order to confirm the interaction between c-Abl and TDP-43, we performed a pulldown assay with SH-SY5Y cells co-transfected with Flag-tagged c-Abl and EGFP-tagged TDP-43. TDP-43 was efficiently co-precipitated with c-Abl, supporting that c-Abl binds to TDP-43 (Figure 2A). Most of the TDP-43 N-terminal domain (PDB ID: 5MDI) consists of a β-strand structure. c-Abl (PDB ID: 2E2B) forms a typical kinase structure where two lobes composed of a/b folds meet with the active site in the middle. Once the structures were docked with each other, the β-strand portion of TDP-43 seemed to be closely coupled to the loop above the active site of c-Abl (Figure 2B). At the bond between TDP-43 and c-Abl, a charged interaction with a hydrogen bond centered on Y43 of TDP-43, a potential phosphorylation site by c-Abl, was observed. It is predicted that Y43 of TDP-43 could be a hydrogen bond with Q300 of c-Abl. At the edge of the bond, hydrophobic contact was mainly observed. According to this model, there is a possibility that Y43 of TDP-43 can directly or indirectly bind with Q300 and E316 of c-Abl, and these residues are located near the active site of c-Abl.

### 3.3. c-Abl Affects Cytoplasmic Location of TDP-43 Depending on Kinase Activity

Previous fluorescent imaging analyses showed that TDP-43 is mainly localized in the nucleus due to a bipartite nucleus localization signal (NLS) sequence in the N-terminal domain upstream of the first RNA recognition motif (RRM) [30]. As such, we determined the effect of c-Abl on the localization of TDP-43. Immunofluorescence analysis revealed that the cytoplasmic localized TDP-43 was increased in the cells expressing a constitutively active form of c-Abl, but not in the cells expressing c-Abl kinase dead (KD) mutant (Figure 3A,B). To further confirm the altered location of TDP-43 by c-Abl, the cells expressing the kinase active form (KA) or c-Abl KD were fractionated for separating the nucleus and cytoplasmic fraction. The levels of cytoplasmic TDP-43 were lower than the levels of nuclear TDP-43 and c-Abl significantly retained the cytoplasmic localization of TDP-43, which depended on its kinase activity (Figure 3C,D).

### 3.4. Tyrosine 43 Phosphorylation by c-Abl Promotes Stress Granule Formation by Cytoplasmic TDP-43

Since the accumulation of cytoplasmic TDP-43 has been physiologically associated with stress granules under the oxidative and osmotic stress [31], we tested whether the accumulation of cytoplasmic of TDP-43 induced by c-Abl regulates the formation of stress granule. To determine the number of stress granules, we stained the G3BP1, which is required for normal stress granule assembly, in the SH-SY5Y cells expressing either TDP-43 WT or phosphor-mimetic mutant (Y43E and Y155E). TDP-43 Y43E mutant increased the number of G3BP1-positive stress granules compared with TDP-43 WT and Y155E mutant (Figure 4A,B). The accumulation of cytoplasmic TDP-43 was also observed in primary cortical neurons with lenti-viral transduction of c-Abl KA, but not by c-Abl KD (Figure 4C). Moreover, cortical neurons expressing TDP-43 Y43E phosphor-mimetic mutant showed the cytoplasmic inclusion of TDP-43 in the cell body and neurites compared to TDP-43 WT expressing neurons (Figure 4D).

### 3.5. c-Abl–Mediated Aggregation of TDP-43 Induces Neuronal Cell Death

It has been proposed that the mislocalization of TDP-43 could nucleate the cytoplasmic inclusions at the stress granules as a site of initial TDP-43 aggregation [32]. To assess the effect of c-Abl–induced phosphorylation on formation of TDP-43 aggregation, primary cortical neurons cultured from WT and c-Abl knockout (KO) mice were transduced with lenti-virus expressing TDP-43 WT or phosphor-mimetic Y43E mutant. After 7 days post-infection, cellular proteins were sequentially extracted in Triton X-100 and urea containing buffers. Unlike the levels of soluble TDP-43, insoluble TDP-43 full-length and CTFs in urea fraction were increased in TDP-43 WT overexpressed neurons. The accumulation was further exacerbated in TDP-43 Y43E overexpressed neurons (Figure 5A). However, in c-Abl KO neurons, the induction of TDP-43 WT failed to accumulate insoluble TDP-43 in urea fraction, whereas the induction of TDP-43 Y43E still sustained the accumulation of insoluble TDP-43 in urea fraction (Figure 5A). Notably, the truncated forms of TDP-43, which may be involved in the multiple pathologies of ALS [33], were also observed in urea fraction from TDP-43 augmented neurons (Figure 5A). Moreover, the levels of truncated TDP-43 depended on the phosphorylated status of TDP-43. These observations support that the accumulation of insoluble TDP-43 is mediated by the function of c-Abl kinase. Consistent with these findings, the levels of insoluble TDP-43 were also observed in both c-Abl WT and KO neurons expressing TDP-43 Y43E mutant, but not in c-Abl KO neurons expressing TDP-43 WT (Figure 5B). Based on these observations, we then determined whether the phosphorylation of TDP-43 by c-Abl affects neuronal death. Neuronal cell death was assessed in primary cortical neurons transduced with either lenti-TDP-43 WT, Y43E or phosphor-deficient Y43F mutant viruses. The fluorescent TUNEL staining showed that the augmentation of TDP-43 WT increased the percentage of dead neurons compared to cortical neurons expressing the control vector (Figure 5C,D). The neuronal cell death was exacerbated in cortical neurons with TDP-43 Y43E, but not Y43F mutant. Taken together, these data suggest that c-Abl–induced TDP-43 phosphorylation at Y43 residue decreases solubility and increases aggregation and truncation, which leads to neuronal cell death.

## 4. Discussion

Non-receptor tyrosine kinase, c-Abl can be activated by several stimuli such as DNA damage and oxidative stress [34,35]. It has been reported that c-Abl is active in multiple neurodegenerative disorders, including Alzheimer’s disease, Parkinson’s disease, and ALS [16,29,36]. Accumulating evidence support that c-Abl inhibition could be one of therapeutic options to mitigate the pathology of ALS [17,18,19], suggesting that c-Abl can closely regulate the pathological downstream processes in ALS. However, c-Abl pathway in ALS has been not fully understood yet.

This study investigated the role of c-Abl in the mislocalization and aggregation of TDP-43 protein by post-translational modification. Proteomics analysis for identifying potential phosphorylation site and site-directed mutagenesis experiment revealed that Y43 is the site of phosphorylation of TDP-43 by c-Abl (Figure 1). In silico docking assay also showed that the β-strand containing Y43 of TDP-43 closely projects to the active site of c-Abl, suggesting kinase domain of c-Abl easily accesses and transfers the phosphate to the Y43 of TDP-43 (Figure 2), which is a residue that has not been reported previously as a phosphorylation site. Previous studies have investigated the role of phosphorylated TDP-43 in regulating stress granule formation, RNA binding, aggregation and neuronal toxicity in the pathogenesis of ALS [37]. Casein kinase 1 and 2 (CK1 and CK2) were identified as kinases that phosphorylate TDP-43 at multiple sites in the carboxy-terminal domain, leading to the formation of insoluble oligomers and filaments of TDP-43 [12]. In addition to CK1 and CK2, cell division cycle 7 (CDC7) also directly phosphorylated TDP-43 at S409/410, which are pathological residues [38]. It has been reported that phosphorylated TDP-43 aggregates were observed along with CDC7, Tau-tubulin kinase 1 and 2 (TTBK1 and TTBK2) in frontal cortex sections from ALS and frontotemporal lobar degeneration (FTLD-TDP) cases, and TTBK1/2 also directly phosphorylated TDP-43 at S409/410 residues in vitro [39]. Not only the importance of the carboxy-terminal region of TDP-43, but also amino-terminal domain (NTD) composed of residues 1–77 has been demonstrated to mediate aberrant aggregation of TDP-43 as well [40]. Consistent with this notion, our data support that post-translation modification of NTD of TDP-43 is important for regulating TDP-43 proteinopathy.

It has been well established that the mislocalization of TDP-43 is tightly associated in the pathogenesis of ALS [7]. We found that c-Abl–mediated TDP-43 phosphorylation increases the cytoplasmic levels of TDP-43 depending on its kinase activity (Figure 3). It speculates that c-Abl may promote translocation of TDP-43 from the nucleus to the cytoplasm via Y43 phosphorylation. Because β-strand containing Y43 is near to NLS of TDP-43 composed of residues 82–98, Y43 phosphorylation of TDP-43 by c-Abl may also interfere the assembly of importin complex for nuclear transport. Both possibilities may result in increasing the retained cytoplasmic TDP-43, but further studies are needed to clarify the function of c-Abl for TDP-43 localization. Nonetheless, the mislocalization of TDP-43 by c-Abl was not shown in cells expressing kinase-dead mutant of c-Abl, suggesting the importance of c-Abl enzymatic activity in the cytoplasmic location of TDP-43. We also found that c-Abl–mediated TDP-43 phosphorylation promotes to form insoluble 35 kDa and 25 kDa CTFs (Figure 5), which lose the NLS [41]. Since CTFs seem to co-aggregate with full-length TDP-43, this process may affect the local of TDP-43 in the presence of c-Abl activation. Further studies are needed to determine the detailed underlying mechanisms generating of TDP-43 CTFs by c-Abl for understanding the role of c-Abl in the pathogenesis of ALS. One of plausible explanations is that a range of cysteine proteases including caspases and calpains, which are responsible for processing of TDP-43, has been known to be activated in the condition of stress involved in c-Abl activation [42,43]. It speculates that the translocation of TDP-43 by c-Abl and the processing of TDP-43 by proteases occur simultaneously at the same condition. It is plausible that c-Abl may contribute to the cleavage of TDP-43 by proteases either exposing the residue due to Y43 phosphorylation or directly altering protease activities. Previous studies demonstrated that c-Abl could affect the activation of caspase-3 regulating the autocleavage of caspase-9 via direct binding [44]. Moreover, it has been reported that the N-terminal end of TDP-43 is a cleavage-prone area and more accessible to proteases [45]. It is likely that phosphorylated TDP-43 at Y43 has less founded in the post-mortem ALS and FTLD-TDP brain tissues compared to phosphorylated TDP-43 in the carboxy-terminal domain.

Our observations were based on the augmentation of active c-Abl in cell models. This limitation should be strengthened by unraveling the molecular mechanisms of c-Abl activation in the pathogenesis of ALS. Previous studies reported that genetic deletion and pharmacological inhibition of c-Abl alleviate the degeneration of motor neurons derived from patients with familial ALS harboring mutations in SOD1, C9orf72, TARDBP [17]. Genetic interaction between c-Abl and ALS-causing genes would be an interesting research area to investigate the pathological mechanisms in the disease onset. In this study, we demonstrated c-Abl is the novel kinase targeting TDP-43 at Y43 in the NTD. This post-translational modification of TDP-43 promoted cytoplasmic localization and accumulation of TDP-43 leading to neuronal cell death. Our data support the importance of c-Abl in the pathogenesis of ALS.

## Figures and Tables

**Figure 1 cells-11-03972-f001:**
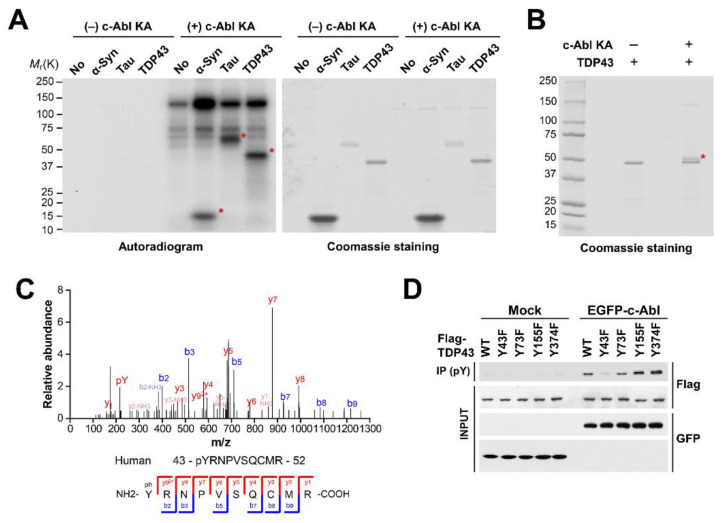
c-Abl phosphorylates TDP-43 at tyrosine 43. (**A**) in vitro kinase assay with recombinant Kinase active form of c-Abl (c-Abl KA) and TDP-43. (**B**) Non-radioactive kinase assay adding cold ATP. c-Abl–phosphorylated TDP-43 indicated by asterisk. (**C**) MS/MS spectrum of the phosphorylated peptide fragment of c-Abl–phosphorylated TDP-43. (**D**) Co-immunoprecipitation experiments in SH-SY5Y cells co-transfected with EGFP-c-Abl and phospho-deficient mutant forms of TDP-43.

**Figure 2 cells-11-03972-f002:**
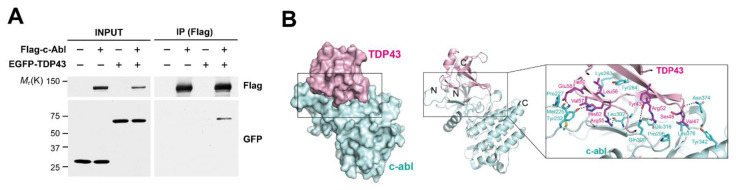
c-Abl interacts with TDP-43. (**A**) Co-immunoprecipitation experiments in SH-SY5Y cells co-transfected with Flag-c-Abl and EGFP-TDP-43. (**B**) The docking model for TDP-43 (red) interacting with c-Abl (cyan). Potential hydrogen bonds are shown as dashed lines.

**Figure 3 cells-11-03972-f003:**
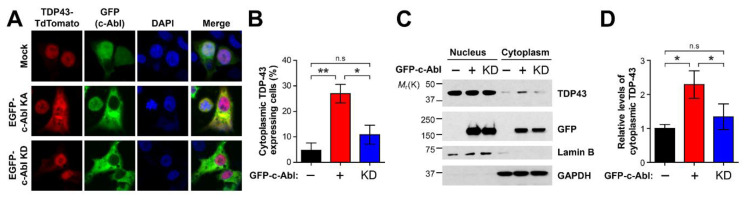
c-Abl increases cytoplasmic distribution of endogenous TDP-43. (**A**) Immunocytochemical analyses of c-Abl and TDP-43. GFP-c-Abl KA (green) or GFP-c-Abl KD and TDP-43-TdTomato (red) were co-expressed in SH-SY5Y cells. (**B**) Quantification of the percentage of cells expressing cytoplasmic TDP-43 (n = 4–12). (**C**) Expression of kinase active (KA) form of c-Abl increases the accumulation of TDP-43 in the cytoplasm, but not in expression of kinase dead (KD) form of c-Abl. (**D**) Quantification of the levels of cytoplasmic TDP-43 in the c-Abl KA and KD–expressing cells (n = 3). Statistical significance was determined by using one-way ANOVA with Tukey’s correction; * *p* < 0.05, ** *p* < 0.01, n.s. = non-significance.

**Figure 4 cells-11-03972-f004:**
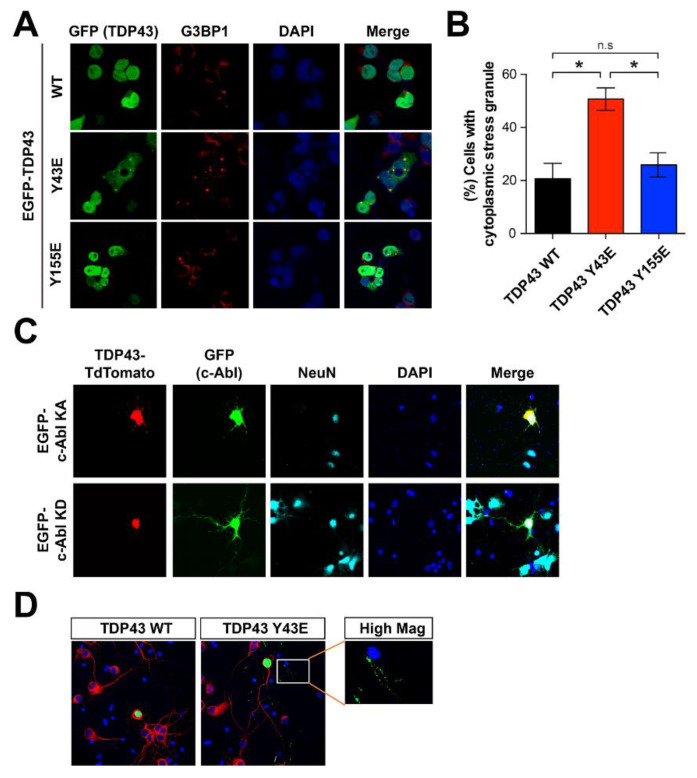
Y43 residue of TDP-43 associates with the TDP-43 pathology. (**A**) Immunostaining with G3BP1 antibody that can detect cytoplasmic stress granules in SH-SY5Y cells transfected with GFP-WT TDP-43, GFP-phospho-mimic forms of Y43E and Y155E TDP-43. (**B**) Quantification of the percentage of cells with cytoplasmic stress granules (n = 3). Statistical significance was determined by using one-way ANOVA with Tukey’s correction; * *p* < 0.05, n.s. = non-significance. (**C**) Increased localization of TDP-43 at neurite granules by c-Abl KA co-expression in cultured cortical neurons. TDP-43 TdTomato and EGFP-c-Abl were expressed in the cultured cortical neurons and followed by immunocytochemistry. (**D**) Phospho-mimic form of TDP-43 Y43E showed increased neurite granule expression than TDP-43 WT in primary cultured cortical neurons. EGFP-TDP-43 WT and Y43E were expressed alone in the cultured cortical neurons and followed by immunocytochemical alayses.

**Figure 5 cells-11-03972-f005:**
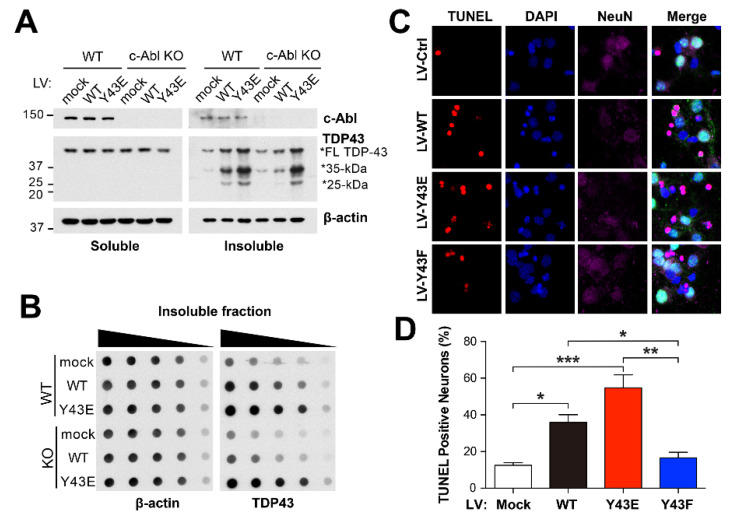
TDP-43 Y43E increases aggregates and neuronal cell death in primary cortical neurons. (**A**) RIPA/urea-fractionated cell lysates with RIPA (Soluble) and urea (Insoluble) buffers from c-Abl WT and c-Abl KO neurons were examined using TDP-43 antibody (**B**) Urea-fractionated cell lysates from c-Abl WT and c-Abl KO were collected and subjected to dot blotting by TDP-43 antibody. (**C**,**D**) TUNEL staining of the cortical neurons shows neurotoxicity of Lenti-viral induced overexpression of indicated TDP-43 plasmids. The percentage of TUNEL-positive neurons showing apoptotic nuclei is quantified and shown in (**D**). Statistical significance was determined by using one-way ANOVA with Tukey’s correction; * *p* < 0.05, ** *p* < 0.01, *** *p* < 0.001.

## Data Availability

All data associated with this study are available in the main text or are available through the corresponding author upon request.

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
