# Peer review of "c-Abl Regulates the Pathological Deposition of TDP-43 via Tyrosine 43 Phosphorylation"

_cells, 2022, doi:10.3390/cells11243972_

Round 1

Reviewer 1 Report

The association of abnormal aggregation of pathological TDP43 with ALS and other neurodegenerative diseases, as well the role of c-Abl as an important pathogenic mediator in NDD is well recognized. It is well documented that c-Abl inhibitors, such as nilotinib, HSP90, PI3K, Akt, mTOR, and MAPK inhibitors reduce the TDP25 levels and are currently intensively investigated as  promising strategy for the treatment of ALS. However, the manuscript by Lee et al. contribute to better understanding of ALS pathogenesis. It is well written and easy to follow, well referenced. The methodology is explained in details, study findings are discussed appropriately and limitations of the study are also highlighted. In my opinion, this manuscript may be of clinical interest.

Author Response

We thank the referee for indicating that our work has a novel, high impact, and succinct summary of our major findings.

Reviewer 2 Report

The authors have identified the position Y43 of the ALS-associated protein TDP-43 as substrate for the kinase c-Abl. They present sound results spanning from in vitro assays to experiments performed  in primary cell cultures. 

I enjoyed reading this work and I think the results are quite convincing. However, I would recommend few minor correction to render the data stronger.

1) Since the introduction and throughout the manuscript, the authors imply that the phosphorylation of TDP-43 byc-Abl occurs in the nucleus. I can see why they assume as such, but I did not find any results proving this. It is true that c-Abl has three NLSs, but it is also reported to be present in the cytosol (e.g.: DOIs 10.1073/pnas.95.13.7457 and 10.1042/BC20080020). Also TDP-43 is reported to shuffle in and out the nucleus when it acts as RNA transporter. Without evidence of nuclear phopshorylation, I would refrain from presenting this as fact. The author should speculate their resoning in the discussion and avoid making claims not supported by results.

2) In the invitro  kinase assay, have the authors ensured that the proteins they purchased from external sources (TDP-43 and tau) are indeed in their monomeric form at the moment in which the assays were perfomed? They are highly aggregation-prone proteins in vitro and I would not be surprised to see pre-formed aggregates. An analytical FPLC, a native gel or a DLS analysis would answer this question easily. Moreover, it would actually be interesting to know whether TDP-43 phosphorylation only occurs on the soluble protein or whether it is also possible on insoluble aggregates.

Within these assays, I would recommend to add a substrate as negative control (e.i., a protein known NOT to be a substrate for c-Abl).

3) It is not clear how the authors induced the formation of stress granules in SH-SY5Y cells. They report to use G3BP1 to track them but there is no mention of the protocol used to cause them. If they did not stress the cells, how can they say for sure that those puncti are indeed SGs? If it has not been done, I recommend to repeat the imaging after active induction of SG formation. If it has been done, please provide an appropriate paragraph in the M&M section of the manuscript.

4) The authors report to visualise, on their WBs, C-terminal fragments of TDP-43 of ca. 25 and 35 kDa. Have the authors checked that the anti-TDP-43 antibody they are using can indeed recognise C-terminal fragments? The product nnumber of the antibody is not reported in the M&M so I could not check this myself.

5) In the last paragraph of the results, the authors report to see "fiblrils" of TDP-43 in c-Abl WT and KO neurons expressing TDP-43 Y43E mutant. I am curious to know how they can call the structures that they see "fibrils" and what is the definition of "fibrils" for them. Do they refer to cross-beta-containing structures? If so, have they maybe stained the cells with an amyloid intercalator, such as Thioflavin S?

6) The work would truly benefict from more substantial statistical analysis from the imaging results. I would be more convinced about this result if a proper analysis was done on mutiple images, counting a total of at least 1000 cells. I could suggest, for example, pre-set macros for ImageJ such as Aggrecount (https://aggrecount.github.io/), which would make this job much faster and easier.
